# Fruit Volume and Leaf-Area Determination of Cabbage by a Neural-Network-Based Instance Segmentation for Different Growth Stages

**DOI:** 10.3390/s23010129

**Published:** 2022-12-23

**Authors:** Nils Lüling, David Reiser, Jonas Straub, Alexander Stana, Hans W. Griepentrog

**Affiliations:** Department of Technology in Crop Production, University of Hohenheim, 70599 Stuttgart, Germany

**Keywords:** vegetation monitoring, leaf area, instance segmentation, cabbage

## Abstract

Fruit volume and leaf area are important indicators to draw conclusions about the growth condition of the plant. However, the current methods of manual measuring morphological plant properties, such as fruit volume and leaf area, are time consuming and mainly destructive. In this research, an image-based approach for the non-destructive determination of fruit volume and for the total leaf area over three growth stages for cabbage (*brassica oleracea*) is presented. For this purpose, a mask-region-based convolutional neural network (Mask R-CNN) based on a Resnet-101 backbone was trained to segment the cabbage fruit from the leaves and assign it to the corresponding plant. Combining the segmentation results with depth information through a structure-from-motion approach, the leaf length of single leaves, as well as the fruit volume of individual plants, can be calculated. The results indicated that even with a single RGB camera, the developed methods provided a mean accuracy of fruit volume of 87% and a mean accuracy of total leaf area of 90.9%, over three growth stages on an individual plant level.

## 1. Introduction

The importance of the sensor-based analysis of agricultural crops increases as more and more processes on farms become automated. Plant growth and plant stage are decisive parameters for planning crop protection and fertilization treatments. Furthermore, information about the predicted yield and the quality of the crops are useful for harvest logistics and product sales [1]. With the decreasing costs of cameras and the further development in computer vision and artificial intelligence, manual field measurements can be substituted and even automated [2]. In this way, plants can be captured and simulated in a non-contact and non-destructive way. The creation of a so-called digital twin gives the farmer or breeder completely new possibilities for analyzing and planning applications [3]. By using artificial intelligence-based methods for the direct acquisition of individual plant data, the step from a precision farming system to a smart farming system is taken. The use of automated processes for remote crop monitoring, compared to manual measurements in the field, is one of the key points of smart farming systems [4].

For example, time-consuming and error-prone methods for determining nitrogen availability in the soil, such as the N-min method [5], could be dispensed since studies have shown a correlation between nitrogen availability, leaf length, and fruit volume [6,7]. Since a leaf area (LA) calculation is very labor-intensive and a fruit volume measurement usually leads to damage of the plant, these correlations have only be used insufficiently in practice so far. Plant-specific analyses by artificial intelligence could lead to more cost-effective and sustainable agriculture, especially for crops with a high fertilization input, such as cabbage [8].

A widely used approach for calculating leaf area is to calculate the leaf area index (LAI) [9,10]. It provides information about the degree of cover, which can be used to indirectly determine the leaf area. LAI can be negatively affected by weed cover in the field as this results in an incorrect crop cover ratio. For this purpose, thresh-holding methods are used, which involve a high post-processing effort, as well as artificial-intelligence-based methods with less post-processing effort [11].

For a more precise and direct measurement of the leaf area, leaf area meters can be used [12]. By scanning individual leaves, an exact statement can be made about the total leaf area [13]. This method, however, is very time consuming and will cause damage to crops such as cabbage due to its sensitive leaves. The most common method of measuring leaf area and crop volume is probably the manual measurement of plant parameters such as leaf length, leaf width, and crop diameter with a measuring tape [14]. The measured parameters are then used with a formula to determine the fruit volume, for example. The research results are published attempting to create mathematical models to describe the actual leaf area (LA) for different kind of plants (Equation (1)). The leaf length (L), leaf width (W), and calculation constant (A) provide the basis for these calculations [15,16,17].
(1)LA= L∗ W∗A

With the increasing possibilities of image analysis through artificial intelligence and the decreasing costs of imaging hardware, new possibilities regarding the automated and direct measurement of plant parameters such as leaf area or fruit volume are increasingly emerging. Shabani et al. [12] already show the applicability of neural networks for leaf-area calculation, including leaf length and width under isolated conditions. However, as the method is presented, the leaves must either be removed or fully visible and therefore cannot be used for on-field operations.

A non-destructive approach to automated leaf-area calculation is shown by Itakura et al. [18]. The leaf area can be determined with high accuracy using structure from motion (SfM) depth information and a water shed algorithm, for instance, segmentation. However, the analysis of this method only took place with very small plants with 3 to 11 leaves under laboratory conditions. Current approaches of leaf-area determination still have major problems with overlapping leaves. This makes on-field applications difficult to implement as a lot of leaf area is overlapped, especially in field vegetables. By incorporating the specific plant morphology and including only visible plant characteristics on the leaf-area calculation, a more robust approach could be implemented. As a result, the approach loses accuracy but gains reliability, which is more crucial for a farmer in practical use.

Approaches to leaf-area calculation are usually very plant-specific as the morphological parameters of the plant are crucial for the calculation. Different approaches such as Masuda’s [19] leaf-area calculation for tomato plants using the semantic segmentation of point clouds or the growth-monitoring method of lettuce by Zhang et al. [20] are mostly difficult to adapt to cabbage plants but are a good example of what information is needed for total leaf-area calculation across varieties. There are also many differences in the calculation of fruit volume due to the morphological influence depending on the variety. The shape and visibility of the fruit are decisive parameters for the calculation of the fruit volume [21]. When calculating the leaf area as well as the fruit volume, two decisive pieces of information, independent of the morphology of the plant, become apparent, which are necessary for a determination. These are, on the one hand, information about the shape and position of the object and, on the other hand, the detection or segmentation of the object.

There are a variety of different options for segmenting fruits or leaves in images or point clouds. These include classical computer vision approaches using geometric features [22] or using watershed methods [23], as well as color and depth image-based segmentation approaches [24]. However, due to their good adaptability to variable environmental scenarios and associated high accuracy, artificial-intelligence-based approaches show the most promising results [25]. With the rapid development of neural-network-based image processing in recent years, a wide range of information can be processed from camera images through targeted training of these networks [26]. The basis of this image processing technique is convolutional neural networks (CNNs), which can be used to classify images [27]. Object-detection networks such as the regions with CNN features method (R-CNN) or the you-only-look-once (YOLO) network locate and classify an object in the image [28,29]. To obtain information at a pixel level, semantic segmentations are applied that assign each pixel in the image to a possible class. Fully convolutional neural networks (FCNs), among others, are used for these segmentations [30]. The instance segmentation by a Mask R-CNN is a further development of the semantic segmentation and combines the object detection of the R-CNN with the segmentation of a FCN [31]. By using the leaf segmentation challenge (LSC) datasets, Xu et al. [25] demonstrates the effectiveness of a Mask R-CNN in the instance segmentation of plants, compared to other methods. However, an instance segmentation network such as a Mask R-CNN can only provide two-dimensional information about the location and segmentation area of the objects in an image.

In order to be able to make statements about the volume or position of fruits or leaves, information about the shape and position is a prerequisite. To record plant parameters such as leaf area or fruit volume, sensor-based depth information can be used, giving the two-dimensional masks of the instance segmentation a three-dimensional shape. Attempts using different kinds of depth sensors [14,32] were presented, to obtain information about the LA. Depending on the application, there are several factors influencing the choice of sensor technology for depth-image acquisition. The decisive factors are the required resolution of the depth image, the costs, and the distance between the sensor and the object. For a simple application with high resolution at a short distance, one common approach is the SfM approach, which can calculate depth information without additional sensor technology by motion in the camera image [33,34].

Due to its large and highly overlapping leaves, cabbage is a complex plant to analyze. Especially in late growth stages, both the cabbage head and a large part of the cabbage leaves are occluded. For a perfect analysis of the plant, not only the visible leaf area is of interest, the total leaf area is too [35]. For an automated recording of the individual total leaf area and fruit volume, plant parameters such as the leaf-attachment point and leaf end point must be captured. So far, no research has been published that automatically determines the total leaf area and the fruit volume for different growth stages of cabbage using camera-based images.

The aim of this paper is to describe a non-destructive approach to measure the total leaf area and fruit volume over multiple growth stages of individual cabbage plants. For this purpose, the cabbage head, the individual leaves, and the entire plant should be segmented by a neural-network-based instance segmentation from color images (RGB). By merging the two-dimensional segmentation information with the three-dimensional information of the structure-from-motion (SfM) approach, the cabbage fruit volume as well as the total leaf area, including the non-visible leaf area, can be determined.

## 2. Materials and Methods

### 2.1. Dataset Recording and Ground-Truth Estimation

A dataset of 600 images was recorded at location in Stuttgart, Germany at three different growth stages. All three growth stages were in macro-stage four (the development of vegetative plant parts) with no fruit yet visible in the first stage. The stages can be assigned to BBCH stages 41, 45, and 48 [36].

A standard GoPro camera (Hero 7, GoPro Inc., San Mateo, CA, USA) was used to capture the training images from a vertical perspective. There were always natural-light conditions with no shading of the images or artificial lighting. The camera was moved across the rows at a recording frequency of 60 Hz and a resolution of 1920 × 1440, with a constant height of 900 mm and a speed of 1 m/s to produce high-quality depth images. For the field operation, the camera system is intended to be attached to the robot platform Phoenix [37]. Due to its caterpillar tracks and the generally good field conditions in vegetable cultivation, the platform can ensure relatively smooth camera movement. The cabbage was a storage cabbage of the cultivar Storidor, grown with a row width of 600 mm and a planting distance of 600 mm. As no herbicides were used on the analyzed bedding parcel, there was high weed occurrence during the recordings (Figure 1c).

Ten randomly picked cabbage plants of each growth stage were harvested and measured to validate the calculated fruit volumes and leaf areas. To determine the fruit volume, the circumference of the cabbage head was measured. The metric volume (V) was estimated as
(2)$$V\;=\frac{4}{3}{*\;\pi *\mathrm{radius}}^{3}$$

This calculated volume fits the actual head volume with a correlation coefficient of 0.88 [38] by assuming a spherical shape. To determine the leaf area, all leaves were removed and singled on a plane, and photos were taken. By segmenting the leaves, the leaf area was recorded in pixels. To convert the pixels into metric units, a reference object of a known size was placed on the same plane as the leaves [39].

### 2.2. Depth-Image Calculation

Due to its cost-effectiveness and versatility, depth-information acquisition using an SfM approach is widely used for agricultural applications [14,40]. The linear movement of the camera at a constant height and speed, the short distance between the camera and the object, and the static environment offer good conditions for an SfM method. The high recording frequency of the camera makes it possible to select image pairs with almost any overlap, which allows for high flexibility in the evaluation.

Therefore, in this work, an RGB camera, an SfM, and an MVS (multi-view stereo) algorithm were used to generate depth images (Figure 2) for subsequent 3D evaluation. SfM was used to determine the camera positions and orientations along the plant row, as well as to determine the camera geometry. The first step is to determine key points in two images. If the same key points can be found, they can be used as tie points, and thus the relative orientation between the images can be determined. Next, successively more images are added to the image block, with their respective tie points. After adding additional images, a bundle block adjustment is performed so that the overall errors are as small as possible [41]. In this approach, the scaling is done using a scale bar, which serves as a reference object and thus gives the subsequently generated depth images the correct dimension. The actual generation of the depth images is then done with an MVS algorithm in order to obtain high-resolution depth images for all images. This evaluation and the generation of the depth images were realized using a commercial photogrammetry software [42].

### 2.3. Instance Segmentation

Object detection provides information about the class and the localization of an object by using a bounding box in the image (Figure 3a). Semantic segmentation provides information about the classification of each pixel of a class (Figure 3b). Instance segmentation combines these methods of object detection and semantic segmentation (Figure 3c). An instance segmentation has three outputs for each object to be detected: a classification, a localization, and a segmentation mask [31]. This information is necessary to assign plant parameters to individual plants and to calculate the leaf area.

As an instance segmentation network, a Mask R-CNN with a Resnet-101 backbone was used to detect the cabbages and single leaves of the plants [43]. Figure 4 shows the schematic procedure of the used Mask R-CNN. The Mask R-CNN is an extension of the object detection network Faster R-CNN. In order to generate an additional object mask, the Mask R-CNN has an additional network for segmenting the detected objects compared to the Faster R-CNN. As in the Faster R-CNN, the images are fed into a CNN (Resnet-101), which generates a feature map. The region proposal network (RPN) generates several regions of interest (RoI) from the feature map, which are filtered out using non-max suppression. The RoI align network provides multiple bounding boxes and scales them into a fixed format. The fully convolutional network (FCN) is a segmentation network that generates a binary mask from the feature vector. The fully connected layers provide, as the Faster R-CNN, the object localization and classification [31].

To evaluate the detection and segmentation results, the mean average precision (mAP) of the individual classes was calculated. The calculation of the mAP consists of the calculation of the precision (Equation (3)), the recall (Equation (4)), and the intersection over union (IoU). The precision describes how many of the detected pixels were correctly classified as the ratio between true positives (TP) and false positives (FP). The recall calculation shows the ratio of all correctly segmented pixels (TP) to the missed pixels (False Negative—FN). The IoU is the ratio between the common overlap of the ground-truth segmentation and the generated segmentation to the total segmented area. After the precision and recall metrics have been calculated, the recall values are plotted on the x-axis and the precision values are plotted on the y-axis. The mAP is then determined by the area under the resulting precision and recall curve. For the final calculation of the mAP, the intersection over union (IoU) threshold for a TP is increased step by step from 0.5 to 0.95. The resulting values are averaged to a final mAP value [44].
(3)$$\mathrm{Precision}=\frac{\mathrm{TP}}{\mathrm{TP}+\mathrm{FP}}$$
(4)$$\mathrm{Recall}=\frac{\mathrm{TP}}{\mathrm{TP}+\mathrm{FN}}$$

### 2.4. Volume and Leaf-Area Estimation

For calculating the cabbage volume, the cabbage head area segmented by the Mask R-CNN was used. The number of pixels of the segmented area indicates the cross-section of the cabbage head in pixels (Figure 5a). The volume in pixels of the cabbage can be calculated via the cross-section of the detected area. Since the conversion from pixels to mm depends on the distance to the camera, the detected cabbage cross-section in the pixels was calculated at the height of the cross-section (H) of the cabbage. For this purpose, the radius of the cabbage was subtracted from the height of the center of the cabbage. With the height of the cabbage center and a converting constant (k), it was possible to convert the pixel values to millimeters with:(5)$$r_{\mathrm{mm}}=r_{\mathrm{px}}*\frac{k}{H_{\mathrm{px}}}$$

By knowing the actual cabbage head radius and using Equation (2), the cabbage volume was estimated. To determine the individual leaf area, the leaves of other plants should not be included in the calculation process. By segmenting the entire cabbage plant, the plant-specific leaves were assigned. When more than 80% of a segmented leaf area was in the segmentation area of the whole plant, the leaf was assigned to the plant. 

The difficulty with the camera-based calculation of the leaf area utilizing the leaf length and the leaf width lies in the recording of these parameters. Looking at the cabbage plants in Figure 2, only the edges of the outer leaves are visible. Since the outer leaves are usually longer than the inner ones, the leaf length can be calculated very accurately if the leaf-attachment point is known. The maximum leaf width is hidden for the outer leaves. Therefore, for the following calculations, the leaf length will serve as the basis for calculating the leaf area [17]. In order to be independent of the plant’s growth stage, a method was developed that can be used without an additional orientation by the fruit geometry. To calculate the leaf length, the leaf-attachment point and the leaf endpoints of all leaves are required (Figure 6).

Since the leaf-attachment point is not visible and therefore not detectable by our camera, it has to be estimated. Since all leaves of a plant converge at the stem, the position of the leaves in relation to each other can be used to determine the x and y coordinates of the stem and thereby the approximate leaf-attachment point. This estimation of the leaf-attachment point is done in 3 steps.
First, the depth information is read out for each segmented leaf (Figure 7a). An M-estimator sample consensus algorithm (MSAC), which is a version of the random sample consensus (RANSAC) algorithm, is used to remove outliers and errors in the depth image (Figure 7b). For this purpose, the MSAC algorithm estimates a plane into the point cloud. Points that fall outside of a tolerance range to this plane are removed [45].

2.A plane is then fitted to the points using a multiple linear regression (Figure 8). The plane reflects the orientation and the further course of the leaf up to the leaf-attachment point.

3.By repeating the first two steps for each leaf of a plant, one plane per leaf is calculated (Figure 9a). These planes create a multitude of three-dimensional intersections (Figure 9b). To determine the three-dimensional leaf-attachment point (p), the x, y, and z coordinates of the intersection points in the segmented plant area are averaged. It is assumed that the point where most of the planes converge is the center of the stem where the leaves are attached.

The independence of the leaf-area calculation from the fruit geometry offers the possibility to apply the method described to a variety of growth stages. The next steps in calculating the leaf area is to calculate the leaf endpoint and the transformation from leaf length to leaf area. For this purpose, the three-dimensional coordinate of the segmented leaf (red point), which has the greatest three-dimensional distance to the previously estimated leaf-attachment point (green point), was determined (Figure 10a). The three-dimensional Euclidian distance (L) between the leaf base (p) and the end of the leaf (q) was then calculated (Figure 10b). The curvature of the leaf is not taken into account when calculating the leaf length.

In the last step, the leaf area is calculated using the leaf length (Equation (6)). The individual leaf length is multiplied by the mean value of all leaf lengths of a plant and a correction factor (2.7) for the leaf shape. The mean leaf length and the correction factor of 2.7 are the substitute for the leaf width, which would be necessary for a more precise calculation of the leaf area according to Olfati [17].
(6)$$\mathrm{LA}=\frac{1}{n}*\overset{n}{\underset{i=1}{\sum }}L*L_{i}*2.7\;$$

### 2.5. Software Implementation and Dataset Used

The Mask R-CNN [31] is mainly based on the libraries Tensorflow GPU 1.3.0 [46,47,48]. The calculation of the leaf area and the cabbage volume were implemented with MATLAB (Matlab R2020a, The MathWorks Inc., Natick, MA, USA). The calculation of the depth images and the camera calibration were done with the Agisoft Metashape software (Metashape Professional, Agisoft LLC, St. Petersburg, Russia, 2021). The complete training and the following steps were carried out with a computer with an AMD Ryzen Threadripper 2920X 12-core processor 64 GB RAM and a 24 GB graphics card (GeForce RTX 3090, Nvidia corporate, Santa Clara, CA, USA).

Each training dataset consists of 200 images, and the validation datasets consists of 30 images for each growth stage, with a resolution of 1024 *×* 1024. To avoid overfitting the network, different image augmentation techniques (shift, rotation, scaling, and mirroring) were used, which enlarged the training data set depending on the number of epochs. With 25 epochs, a training data set of 5000 augmented images was created for each growth stage. Furthermore, a transfer-learning approach based on the Microsoft Common Objects in Context (COCO) data set was applied to achieve usable results, even with a comparatively small data set [49]. In a transfer-learning approach, an already trained neural network is further trained with new classes and images. In this way, already similar trained features can be adopted, which reduces the training effort and increases the network accuracy [50]. The training datasets were trained in three classes (Plant, Crop, and Leaf) over 25 epochs, with a stochastic gradient descent, an image batch size of six, and a learning rate of 0.0001.

## 3. Results

Table 1 shows the segmentation accuracies of the cabbage head, the whole plant, and the leaves, over the three growth stages. At growth stage BBCH 41, the cabbage head just started to form and was not visible or detectable (Figure 1a). The trained Mask R-CNN performed well for detecting the cabbage plants and cabbage heads. The deviations in the segmentation accuracy of the leaves are due to the large variations of the leaves in shape, color, and size, which makes precise segmentation by a Mask R-CNN difficult. The deviations in the segmentation of the cabbage head at growth-stage BBCH 45 are due to the size of the cabbage head. As the cabbage head was still very small at this growth stage and was heavily covered by leaves, precise labelling and segmentation were difficult.

The following Table 2, Table 3, Table 4, Table 5 and Table 6 present the results of cabbage fruit volume and leaf-area determination. The results for the measured, calculated, and detected parameters are compared. The measured results were taken from the harvested cabbages and visible leaves that were measured by hand individually. The calculated results show the outcomes of the developed methods, as described in Section 2.4, for cabbage volume and leaf-area calculation with the ground-truth data of the segmentation. The calculated results show the potential in the accuracy of the developed method. The detected results show the outcomes of the developed method for fruit volume and leaf-area calculation using the trained instance segmentation network. The detected results illustrate the applicability and robustness of the methods. The results of the detected parameters refer to the calculated results as these represent the optimum of the method used.

### 3.1. Cabbage Volume

Table 2 and Table 3 show the measured, calculated, and detected results of the cabbage volume. At growth stage BBCH 48, an average cabbage volume of 0.0029 m^3^ was measured. With a standard deviation of 0.00057 m^3^ (19.89%), the measured cabbages show significant variation in their size despite the same planting date and the same growth conditions. An average accuracy for the calculated volume of 87.2 % with a mean value of 0.0027 m^3^ and an accuracy for the detected volume of 86.7 % with a mean value of 0.0025 m^3^ had been achieved at the BBCH stage 48. The deviations in the calculated volume were mainly due to annotation errors of the ground-truth data set. Since leaves often obscure the real cabbage surface, the cabbage outline had to be estimated when creating the ground-truth data. Attempts to fit a sphere into the depth images by using a RANSAC algorithm, in order to be able to dispense with the erroneous annotation data, showed only very poor results as the sphere too often oriented itself to the wrong points.

At growth stage BBCH 45, an average cabbage volume of 0.00092 m^3^ was measured. With a standard deviation of 0.00027 m^3^ (30.06%), the measured cabbages displayed significant variation in terms of their size despite the same planting date and the same growth conditions. Table 3 shows the calculated volume with an average accuracy of volume estimation of 84.9% with a mean value of 0.00083 m^3,^ as well as the detected one with 84.33% and a mean value of 0.00078 m^3^, at growth stage BBCH 45. In this case, a significantly smaller cabbage head volume can be observed due to the earlier growth stage. Furthermore, the calculated ground-truth deviation from the measured value also increases because labelling is made even more difficult with smaller fruit sizes due to the leaves’ even more significant overlapping.

### 3.2. Leaf-Area Calculation

For the calculation of the leaf area, only the leaves visible from the camera perspective were evaluated. For the evaluated cabbage plants, up to 5 leaves were not visible. This may be due to the heavy occlusion of the leaves as well as the high weed pressure that was present in the field. To calculate the correct leaf area, a correction factor could be included to estimate non-visible leaves. However, this problem is only expected in late growth stages, where a large proportion of the leaves overlap. Table 4, Table 5 and Table 6 compare the detected leaf area with the measured visible leaf area and with the leaf area calculated. For calculating the leaf length, the leaf-attachment point was standardized to one point for all leaves. Experiments with individual leaf-attachment points for each leaf, assuming a linear z-progression of the stem at the x-y-coordinate, showed large deviations. Especially in the center of the plant, large deviations were found as the depth information of these leaves does not have a funnel-shaped orientation. The leaves show more of a spherical surface, whereby the corresponding plane would determine a leaf-attachment point that is significantly too high.

With the method created for calculating the leaf area, an average accuracy of 90.9 % from the actual leaf area with a mean value of 0.65 m^2^ was achieved in the latest growth stage (Table 4—BBCH 48). The measured ground-truth values show an average leaf area of 0.68 m^2^ with an average number of leaves of 13.9. The equation for calculating the leaf area was adapted to Olfati [17]. Due to the segmentation results of the Mask R-CNN, an average accuracy of 94.1 % from the calculated value was achieved. The deviations could have been caused by damaged leaves and by the incorrect estimate of the leaf length when the real leaf end was not visible.

Table 5 shows the results of growth stage BBCH 45. At this growth stage, more leaves were visible, which may have been due to the smaller cabbage head. On average, there were 19.4 leaves per plant at this growth stage, with an average area of 0.037 m^2^ per leaf. At growth stage 48, there were on average 13.9 leaves per plant with an average leaf area of 0.049 m^2^ per leaf. The calculated leaf area of 90.3% shows a comparable accuracy to growth stage 48. Comparing the detection accuracy of the two growth stages, growth stage 45 shows a significantly lower accuracy of 84.9%. This is due to the poorer leaf detection. In growth stage 45, a total of 32 leaves were not detected. In comparison, only 12 leaves were not detected in growth stage 48. This may be due to the smaller number and bigger size of the leaves. In addition, growth stage 45 shows a significantly larger standard deviation of the measured leaf area of 20.21%.

Table 6 shows the results of the early growth stage (BBCH 41). At this stage of growth, an average of 17.1 leaves with an average area of 0.02 m^2^ per leaf were visible. The leaf-area calculation achieves a comparable accuracy of 91.7 % to the other two growth stages with a mean leaf area of 0.19 m^2^. Due to the low individual leaf area, with a high number of leaves, a total of 20 leaves were not detected. This leads to an accuracy of 87.6 % of the detected LA.

Both for the fruit volume and the leaf area, the measured values across all growth stages show that despite the same planting date and the same growth conditions, the plants develop significantly differently. This is particularly evident in growth stage 45, where there is a standard deviation of over 30% in fruit size and 20.2 % in leaf area. This shows that individual measurements are important for, e.g., yield determination or plant specific fertilizing, and that generalized volume data for all plants of a growth stage are not accurate.

## 4. Discussion

### 4.1. Experimental Setup

As the cabbage plants were completely destroyed during the measurement of the ground-truth values, only a limited number of cabbage plants could be used during the vegetation period. With the 30 harvested cabbage plants, a total of 504 individual leaves could be measured, which provides a good data basis for the leaf-area calculation. For subsequent research, a significantly larger number of data over more growth stages would be required, especially for the determination of fruit volume, in order to be able to make a statistically valid statement.

### 4.2. Depth Image Calculation

The SfM method offers many advantages for the generation of depth images. Due to the short distance between camera and plant, an SfM method is beneficial because there is no minimum distance for depth imaging, like there is in the case of commercially available RGB-D cameras [51]. By using an SfM method, depth images with higher resolutions can be generated without having to resort to expensive RGB-D cameras. Furthermore, it provides a robust method of depth-image generation as external exposure influences do not affect the quality of the depth image, as is the case with time-of-flight cameras [38]. To ensure good terrain mobility, artificial shading was not used. In order to minimize problems in the detection of the plant due to shadowing, depth information can be used as an additional source of information for the neural network because it is more robust in terms of exposure using the SfM method [52]. Problems in the SfM depth calculation could occur in the field, especially in the border areas, when strong inclinations of the robot lead to a change in height of the camera. These problems could be minimized with the positioning system of the robot. With additional sensors and a static transformation, the exact camera position could be calculated [32]. To avoid the scaling problems of the SfM approach, a stereo camera could also be used to capture the depth information.

### 4.3. Crop Volume Calculation

The results of the cabbage volume calculation of the later growth stage show a higher accuracy, which is due to the better visibility and delimitation of the significantly larger cabbage head. The deviations of the calculated ground-truth values of the volume from the measured values can be explained by two influences. The calculated method is based on the labelled ground-truth images. If the labelling is incorrect, this automatically results in reduced accuracy. In addition, the ground-truth images serve as a training basis for the detection, which leads to poorer results. To increase accuracy, additional information or sensors could be included in the labelling and training process [51]. Another negative influence on the calculated volume lies in the calculation of the depth information. Due to deviations in the recording height caused by demanding field conditions, a depth calculation using an SfM approach can lead to deviations in accuracy. This error influence also negatively affects the leaf-area calculation. These inaccuracies could be improved by additional sensor technology for the exact description of the camera position. As an alternative method, a multi-camera stereo method could also be used, which provides accurate depth information regardless of the camera position and orientation.

Even with the inaccuracies shown and the correlation of 0.88 of a sphere volume to the actual cabbage volume [38], the cabbage volume calculation could be used in practice for vegetation monitoring and harvest management since the advantages such as the non-destructive measurement and the high working speed outweigh the small deviations in accuracy. However, for estimating other cabbage varieties, the development of more adapted geometric methods would be needed.

### 4.4. Leaf-Area Calculation

The results of the leaf-area calculation based on the ground-truth data of an average of 90.9 % confirm the accuracy of the developed method. By additionally recording the leaf width, an even more accurate statement about the leaf area could be made [17]. The significantly lower accuracy of the detected leaf area, especially in growth stages 45 and 41, can be attributed to the high number of leaves with a low leaf area, which are more difficult to detect for a Mask R-CNN. The comparable segmentation accuracies of the leaf areas of the three growth stages show that especially small visible leaf areas were not detected. Comparing the segmentation result of the leaves from growth stage 48 of 0.47 mAP with the result of the leaf area detection of 94.1%, it can be observed that due to the reduction in the leaf-area determination to the leaf length, more minor deviations in the segmentation do not have a substantial effect on the leaf-area calculation. This shows that the detection of the leaves is of higher importance for the subsequent calculation of the leaf area than an exact segmentation of the individual leaves.

To improve the detection results and an implementation into practice, larger data sets would have to be generated and procedures for segmentation improvement would have to be applied [52]. By only including the visible leaves in the calculation process, hidden leaves remain unconsidered. Since these are only a few and contribute only minimally to the photosynthetic performance of the plant, this influence is negligible for practical purposes. Due to the independence of the leaf-area calculation from the fruit geometry, the developed method for leaf-area determination could also be used for other leafy vegetable crops or different cabbage varieties. Despite the inaccuracies shown, this method of leaf-area calculation provides a reasonable basis for further work in vegetation monitoring due to its simplicity and the resulting information. Especially the advantage of speed outweighs here. Information about leaf length or fruit volume can be determined in seconds depending on implementation and the system, which would take a manual worker several minutes to measure.

## 5. Conclusions

A neural-network-based instance segmentation, as well as an SfM method for depth calculation, were used to determine the volume and total leaf area of cabbage plants by using a standard RGB camera over multiple growth stages. The measured standard deviations of fruit size of up to 30% and leaf area of up to 20% show that despite the same growth conditions, plants can differ significantly in their characteristics and that a plant-individual analysis is necessary. By merging the two-dimensional segmentation information from the Mask R-CNN with the three-dimensional information of the SfM approach, several scientific and practice-relevant results could be achieved:With this method, the previous destructive and time-consuming steps of manual measurements can be carried out significantly faster without damaging the plants. It makes direct plant parameter measurement possible, thus enabling the avoidance of measurements of inaccurate indicators such as the LAI, to gain information about the plant development.By determining the fruit cross-section by the Mask R-CNN and assuming a spherical shape, the fruit volume can be determined with 87% accuracy over several growth stages.By determining the leaf-attachment point by the relative position of the leaves and the leaf end point, the leaf length can be determined. The leaf length can then be used to determine with 90.9% accuracy the total leaf area over several growth stages, including overlapping leaf areas.The calculation of the leaf length provides a robust possibility to implement accurate leaf-area determinations, even with deficient segmentation results.By capturing the depth information and segmenting individual plants, cabbages and individual leaves with a minimal sensor setup, a basis for creating a digital twin for smart farming applications is created.

To improve the method shown, more information about the leaf width needs to be collected and approaches to improve detection need to be implemented. This work demonstrates the possibility of creating a digital twin of cabbage, whereby a variety of plant parameters can be recorded remotely. This presents new possibilities for plant-specific analysis in order to plan applications even more efficiently and sustainably.

In addition, future work could analyze the presented leaf-area calculation approach to other cabbage varieties and field vegetables. Due to the independence of leaf-area determination from fruit geometry, this approach could be applied without many adaptations to a wide range of varieties if they have a similar morphology. In the future, it would be advisable to have the images taken by a drone in order to record plant parameters more quickly and without the risk of damage to plants by the robot. The methods developed would only have to be minimally adapted for this purpose.

## Figures and Tables

**Figure 1 sensors-23-00129-f001:**
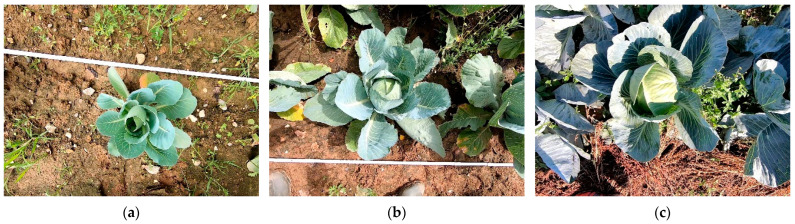
Cabbage plants of BBCH stage 41 (**a**), 45 (**b**), and 48 (**c**).

**Figure 2 sensors-23-00129-f002:**
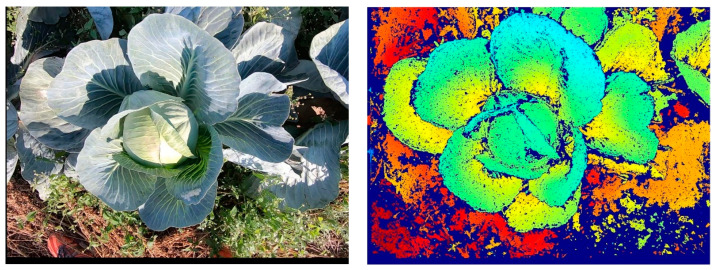
Example for an RGB image with a corresponding depth image (colored).

**Figure 3 sensors-23-00129-f003:**
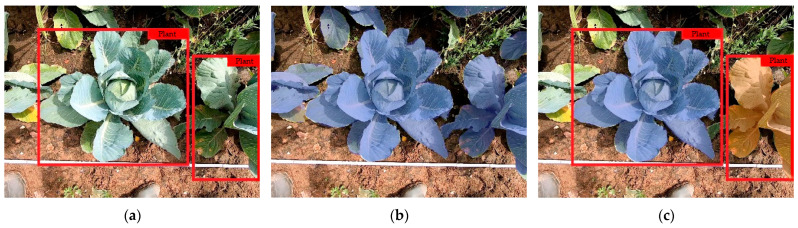
Examples of object detection by a bounding box (**a**), semantic segmentation (**b**), and instance segmentation (**c**).

**Figure 4 sensors-23-00129-f004:**
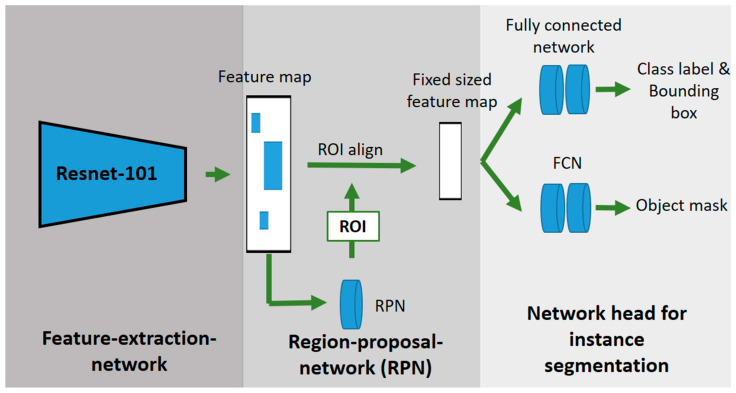
Schematic structure of a Mask-R-CNN-based on He et al. [31].

**Figure 5 sensors-23-00129-f005:**
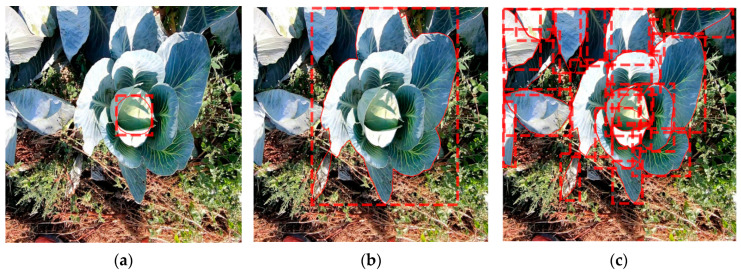
Ground truth instance segmentation masks of the cabbage head (**a**), the whole cabbage plant (**b**) and all cabbage leaves (**c**) at BBCH stage 48.

**Figure 6 sensors-23-00129-f006:**
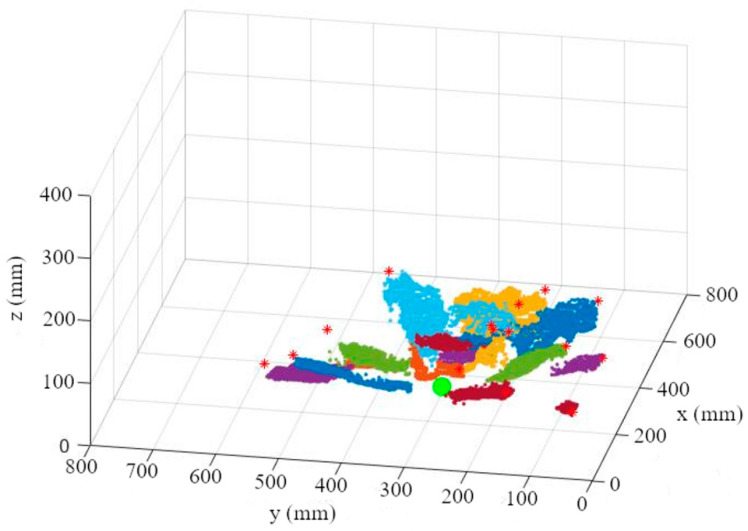
Leaf point cloud with marked leaf-attachment point (big green point) and leaf end-points (*).

**Figure 7 sensors-23-00129-f007:**
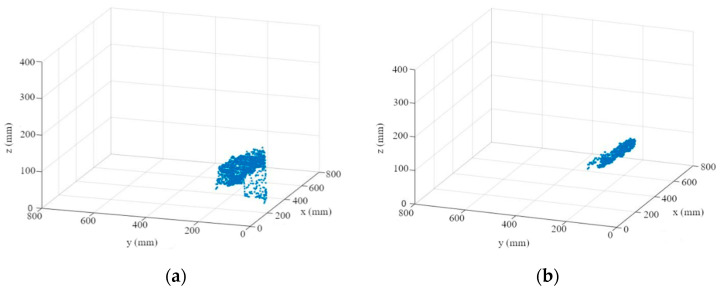
Leaf point cloud (**a**); point cloud after using MSAC algorithm (**b**).

**Figure 8 sensors-23-00129-f008:**
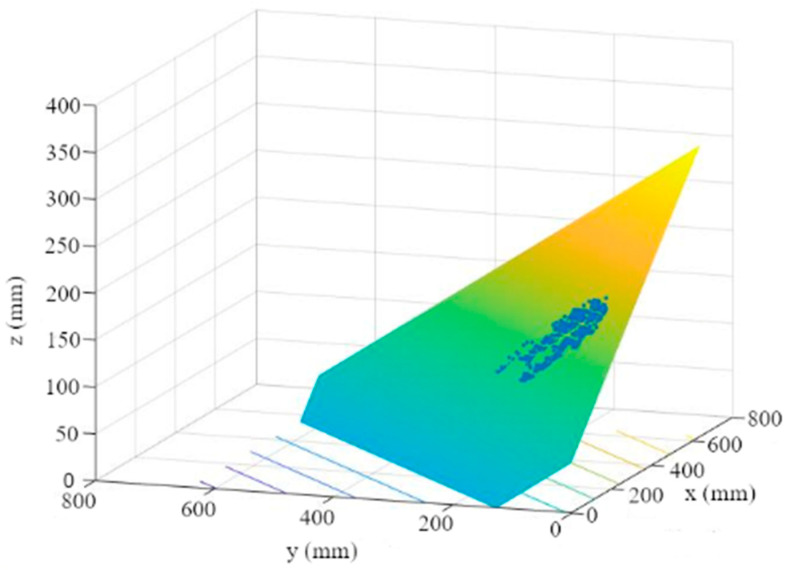
Leaf plane through MSAC point cloud.

**Figure 9 sensors-23-00129-f009:**
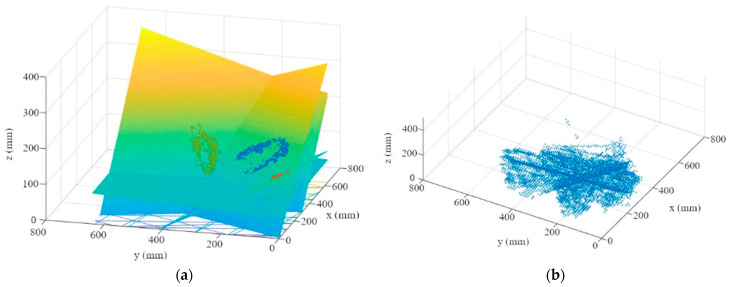
(**a**) Visualization of the first four leaf planes; (**b**) three-dimensional visualization of the plane intersections in the plant area.

**Figure 10 sensors-23-00129-f010:**
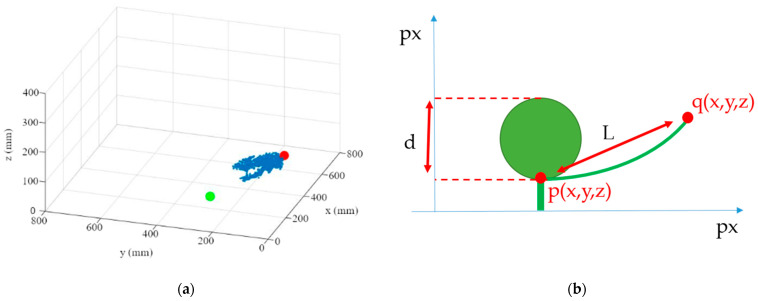
(**a**) Point cloud of the leaf with cabbage center green and leaf endpoint red; (**b**) sketch of the side-view of the cabbage head and leaf with measuring points noted.

**Table 1 sensors-23-00129-t001:** Mean average precision (mAP) of cabbage head, whole plant, and leaves, at different growth stages [32].

Growth Stage (BBCH)	Cabbage Head (mAP)	Whole Plant (mAP)	Leaves (mAP)
41	-	0.82	0.52
45	0.55	0.61	0.51
48	0.81	0.82	0.47

**Table 2 sensors-23-00129-t002:** Comparison of measured volume, calculated volume, and detected volume in regard to the calculated results of BBCH stage 48.

Cabbage Plant (n)	Measured Volume (m^3^)	Calculated Volume (m^3^)	Detected Volume (m^3^)
1	0.0018	0.0021	0.0021
2	0.0034	0.0037	0.0038
3	0.0028	0.0031	0.0034
4	0.0031	0.0029	0.0033
5	0.0032	0.0029	0.0029
6	0.0029	0.0031	0.0023
7	0.0022	0.0017	0.0015
8	0.0034	0.0027	0.0016
9	0.0036	0.0031	0.0025
10	0.0022	0.0018	0.0016
Mean value (m^3^)	0.0029	0.0027	0.0025
Standard deviation (%)	19.89	22.5	30.8
Mean accuracy (%)		87.2	86.7

**Table 3 sensors-23-00129-t003:** Comparison of measured volume, calculated volume, and detected volume in regard to calculated results of BBCH stage 45.

Cabbage Plant (n)	Measured Volume (m^3^)	Calculated Volume (m^3^)	Detected Volume (m^3^)
1	0.00085	0.00089	0.00091
2	0.00089	0.00089	0.00087
3	0.00055	0.00058	0.00051
4	0.00060	0.00081	0.00069
5	0.00125	0.00104	0.00129
6	0.00100	0.00077	0.00065
7	0.00108	0.00110	0.00104
8	0.00050	0.00048	0.00014
9	0.00125	0.00061	0.00054
10	0.00125	0.00112	0.00114
Mean value (m^3^)	0.00092	0.00083	0.00078
Standard deviation (%)	30.06	25.31	41.2
Mean accuracy (%)		84.9	84.33

**Table 4 sensors-23-00129-t004:** BBCH 48: Comparison of measured, calculated (calc.), and detected leaf area and the mean accuracy (MA) of the visible leaves (vis.).

Plant	Leaves Visible Number	Measured LA vis. (m^2^)	Calculated (m^2^)	Leaves Detected number	Detected LA (m^2^)
1	13	0.57	0.58	12	0.51
2	15	0.71	0.73	13	0.66
3	14	0.67	0.68	14	0.68
4	16	0.71	0.89	16	0.90
5	11	0.60	0.58	11	0.59
6	15	0.85	0.65	13	0.56
7	13	0.62	0.62	13	0.63
8	15	0.82	0.74	13	0.62
9	12	0.61	0.55	12	0.54
10	15	0.67	0.59	11	0.50
Mean Value	13.9	0.68	0.65	12.7	0.61
Standard deviation (%)	10.8	12.7	15.12	11.7	17.7
Mean accuracy (%)			90.9		94.1

**Table 5 sensors-23-00129-t005:** BBCH 45: comparison of measured, calculated (calc.), and detected leaf area and the mean accuracy (MA) of the visible leaves (vis.).

Plant	Leaves Visible Number	Measured LA vis. (m^2^)	Calculated (m^2^)	Leaves Detected Number	Detected LA (m^2^)
1	19	0.70	0.65	16	0.54
2	18	0.69	0.61	17	0.56
3	18	0.65	0.72	16	0.70
4	17	0.59	0.68	14	0.55
5	19	0.79	0.75	16	0.72
6	22	0.80	0.80	17	0.60
7	21	0.88	0.93	19	0.82
8	17	0.42	0.49	13	0.39
9	22	0.91	0.81	13	0.48
10	21	0.91	0.80	21	0.78
Mean Value	19.4	0.73	0.72	16.2	0.61
Standard deviation (%)	9.56	20.21	16.24	14.8	20.95
Mean accuracy (%)			90.3		84.9

**Table 6 sensors-23-00129-t006:** BBCH 41: comparison of measured, calculated (calc.), and detected leaf area and the mean accuracy (MA) of the visible leaves (vis.).

Plant	Leaves Visible Number	Measured LA vis. (m^2^)	Calculated (m^2^)	Leaves Detected Number	Detected LA (m^2^)
1	18	0.18	0.16	13	0.09
2	18	0.18	0.18	15	0.16
3	19	0.21	0.22	17	0.20
4	17	0.22	0.24	15	0.22
5	15	0.23	0.19	12	0.22
6	17	0.19	0.17	14	0.15
7	18	0.23	0.20	13	0.19
8	19	0.22	0.25	15	0.22
9	16	0.15	0.15	14	0.16
10	14	0.19	0.18	14	0.19
Mean Value	17.1	0.19	0.19	15.3	18.1
Standard deviation (%)	12.5	9.2	15.9	10.1	21.4
Mean accuracy (%)			91.7		87.6

## Data Availability

The data presented in this study are available on request from the corresponding author.

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
