# Peer review of "Fruit Volume and Leaf-Area Determination of Cabbage by a Neural-Network-Based Instance Segmentation for Different Growth Stages"

_sensors, 2022, doi:10.3390/s23010129_

Round 1

Reviewer 1 Report

The article deals with a topic of using neural networks for agriculture and this topic is very needed. I think that the article should be extended to be published in high impact factor journal. The description of the application is good, but what is lacking is a broader analysis of similar work and situating the paper in a new area of the AI in agriculture. In general, you have a proper academic way of referring and a good language. I really like the application implementation sections.

Comments and suggestions:

1. In the introduction there is only basic information about the issue. It is customary in quality journals to discuss related solutions and outline where improvements are possible. Bring a more in-depth analysis of related solutions and set your research in this context.

2. The conclusion is quite short. Clearly describe in bullet points the scientific and/or application benefits of your solution. Discuss possibilities for further research.

Reviewer 2 Report

In this paper, the fruit volume and leaf area of cabbage were determined by neural network-based instance segmentation for different growth stages. The study is interesting; however, the manuscript in its present form has some weaknesses. The following should be adequately revised to justify a recommendation for publication.

Please provide descriptive statistics of the data. 

Line (245-246): What is the constant value (2.7)? Please clarify.

There are a lot of figures. Therefore, some of them could be combined (e.g., Figures 7, 8, and 9).

Why did you perform the analysis for ten cabbage plants? 

Were there significant differences among the plants selected for analysis?

What are the limitations and future directions of the study?

Round 2

Reviewer 1 Report

The article can be accepted now.